# How Does an Artisan Cooperative Impact Food Perception and Consumer Behaviors? A Rapid Rural Appraisal of Women in East Africa

**DOI:** 10.3390/foods12213956

**Published:** 2023-10-29

**Authors:** Garrett S. Brogan, Kim E. Dooley, Robert Strong, Laxmi Prasanna Kandi

**Affiliations:** Department of Agricultural Leadership, Education, and Communications, Texas A&M University, College Station, TX 77843, USA; garrett.brogan@ag.tamu.edu (G.S.B.); kim.dooley@ag.tamu.edu (K.E.D.); laxmiprasanna.kandi@ag.tamu.edu (L.P.K.)

**Keywords:** culturally viable diets, consumer perception, women’s cooperatives, women’s empowerment

## Abstract

We examined the impact of artisan cooperatives on East African women related to changes in consumer perception and food choice. We sought to understand how artisan cooperatives were impacting women’s food security, diet, and nutrition. To comprehend the lived experiences of women, a qualitative, phenomenological study was conducted within three purposively selected cooperatives in Rwanda and Kenya. Data was collected using Rapid Rural Appraisal (RRA) methodologies with three activities. When given a choice, women did not always eat healthier despite having more money and information about healthy diets. Diets shifted to include more sugar and fruit but less vegetables. Culture and location impacted how they cooked, what food was available to them, and what foods they chose to consume. Women explained how a higher income provided greater agency when buying food. It is crucial to comprehend how culture affects a person’s diet before implementation of nutrition programs. Further research is needed to determine if cooperatives geared toward women’s empowerment and economic sustainability can enhance the nutritional benefits of increased income in culturally relevant ways. This study has implications regarding sustainable development goals with international development programs that utilize locally sourced foods and are culturally viable.

## 1. Introduction

In the heart of East Africa, where diverse cultures collide with a complex tapestry of traditions, a revolutionary movement is emerging within the artisan industry. Artisan cooperatives are developing as social networks seeking to connect local artisans to information, resources, and markets. These cooperatives not only act as a place to provide dignified work but also provide other social benefits such as nutritional education programs. There continues to be interest in the impact of social networks on individuals who have transitioned from informal sectors (before joining a cooperative) to formalized sectors [1,2,3]. This study will explore these social networks’ impact on changes in women artisan members’ diet, nutrition, and food security toward achieving the UNs 17 Sustainable Development Goals (SDG) by 2030. Understanding the factors impacting the health and livelihood of an individual are important issues to study due to the inter-relatedness of poverty reduction with food security [4,5]. Lack of available nutrition education in developing countries creates significant barriers to healthy diets [6]. Nutrition education programs should be examined for potential benefits for women and children [7]. Community-based education programs have been held in conjunction with food security projects and have positively contributed towards improving children and family nutrition [8]. Women involved in these programs still face significant barriers to improving household diets, such as a lack of access to high-quality foods [8]. 

Consumer choices have complex influences and are intricately woven into culture, society, and individual preferences, making attitudes toward many food-related decisions crucial [9]. Culture is often a component left out of nutrition or food security plans [10]. Concerns regarding food safety, biological quality (taste, smell, etc.), diversity, and even the sustainable development of food production have an enormous effect on decisions associated with food [11]. Consumer behavior theory postulates that knowing a product has specific desirable attributes or the desired attributes are inferred based on cues received from credible sources [12]. Subjective norms, perceived behavioral control, and attitudes are predictors of consumers’ behavioral intentions [13]. Ref. [14] reported educational program interventions produced changes in marginalized women’s nutrition behavior. Short-term experiential learning opportunities may be the best approach to encourage nutritional behavior change, increase nutritional knowledge, and lessen health disparities between under-represented and higher socio-economic individuals [14]. 

The purpose of this research study was to explore the impact social networks have on women artisans’ daily food security, diet, and nutritional knowledge. Exploring how consumer choices are impacted by a steady income and local culture is critical knowledge for change agents and funding agencies implementing international development [15]. An artisan cooperative social network provides opportunities for members to gain new knowledge, skills, and resources [16,17]. This new knowledge impacts women’s agency and ability to make food choices for themselves and their families [18,19].The following research objective provides the gestalt of the phenomenological essence of this study:
RQ1. What impacts does membership in an artisan cooperative have on women related to food security, diet, and nutrition? 

### 1.1. Literature Review

Our study aligns with Sustainable Development Goals 1–5, 10, and 11. Of all 17 sustainable development goals, at a minimum, 11 SDGs have indicators aligning with gender paradigms [20]. SDG 1 is focused on ending poverty for everyone around the world and is one of the greatest global challenges [21]. COVID-19 reversed previous efforts towards achieving this goal to end extreme poverty, which is defined as living on less than USD 2.15 per person per day [21]. SDG 2 has an emphasis on creating a world free of hunger [21]. In 2022, 9.2% of the world’s population experienced chronic hunger, which is an increase compared to 2019 [21]. Indian women increased familial food security due to their knowledge of nutritional localized plant and animal species by utilizing local and cultural knowledge to improve food security [22]. Gender equality is the goal of SDG 5, but gender equality also impacts sustainable and prosperous communities [UN]. Urban agriculture practices have been found to positively increase women’s agency [18]. Increased efforts to improve women’s empowerment have increased child nutrition in sub-Saharan Africa [19]. These results imply the spillover effect SDG 5 has on other goals. The limitations and potential of SDG 5 were examined by [23] to understand the goal’s role in improving food security in households. Inequality threatens both the short- and long-term goals of sustainable communities, reducing poverty and increasing economic development (SDG 10) [UN]. Women working in the Thai jewelry supply chain experienced systemic inequality, which was closely linked to the urban–rural divide [24,25]. Women working in the least formalized levels of the gemstone trade faced greater inequalities [24]. 

Cooperatives act to formalize previously informal sectors like agriculture and artisan industries. Cooperatives provide access to markets, information, and production solutions to a wide array of farmers that an individual producer would have more difficulty in attaining on their own [15]. Artisan cooperatives are pivotal in providing members with new revenue sources and a social network [26]. Artisan cooperatives in China utilized their agency to negotiate higher prices for goods and services produced by the cooperative [25]. In an artisan cooperative study in Ecuador, [27] cooperative members experienced higher social capital, economic advancements, and increased skills to become successful producers juxtaposed to non-artisan cooperative members. A Puerto Rican forestry artisan cooperative leveraged its microenterprise capacity to earn more from products and develop business relationships locally when competing in the global marketplace was not an option [28]. Turkish artisan cooperatives created and provided vocational programs to members and production infrastructure to both educate and improve the supply chains for members [29]. 

### 1.2. Theoretical Framework

Social Capital Theory (SCT) is ascribed to several scholars, but it is most strongly associated with Robert D. Putnam, a political scientist and sociologist [30]. SCT has roots in the work of earlier scholars like Pierre Bourdieu, James Coleman, and Nan Lin, who made significant contributions to this concept’s development. SCT focuses on the valuable assets gained from relationships [31]. These include the diversity of capitals, their adaptability, and their reducibility to economic capital [32]. Bourdieu posits that relationships provide sources of potential economic assets because of increased networking and connections. It has been found that communities with strong social networks are better equipped to handle vulnerable situations and complex issues like poverty or natural disasters [33,34]. Thus, this study used SCT to further investigate how social connections found with an artisan cooperative impacted women’s empowerment, wages, and food choices [30]. 

According to SCT, access to resources by gaining adequate earnings provides agency for women to spend more on a wider selection of foods. This might result in a more varied diet with health benefits [35]. These connections also impact food security and diet. Lately, pastoral women, through their social networks, have been found to engage in food-sharing in times of household food shortages [36]. The availability of information may increase people’s awareness of nutritional best practices and better dietary options. Informal women’s social networks have positive impacts on household food security, and these groups act as effective mechanisms, particularly in rural communities, to improve food security [37,38]. Therefore, SCT recognizes that social networks can enable people to make informed decisions, and women who make a good living may have more freedom to make family decisions, such as what to eat [39]. 

Ref. [40] reported in their study on Vietnamese smallholder farmers that connecting and joining the social capital of members to a larger network of farmers improved members’ knowledge of climate change mitigating strategies. A Greek investigation of agricultural trainees found that social capital resulted in farmers improving their knowledge and creating agricultural innovations [41]. Ref. [42] examined social capital’s effect on smallholder farmers’ adoption of agricultural plant and animal biodiversity practices and improved Chinese rural households’ access to credit programs designed to reduce poverty and improve families’ livelihoods [43]. 

Cooperatives are commonly found across agriculture, and there is an increased presence of agriculture cooperatives for women in developing countries [44]. This phenomenon has transitioned to the artisan sector in developing countries. A large percentage of women work in the artisan sector. The products made by these women have unique cultural ties, which has led to the formation of groups of cooperatives of women who are making comparable products [44]. These social connections provide women with different opportunities to expand their skills and productions by collaborating with other women. This study will focus on how these social connections found within these artisan cooperatives impacted food security and changes in their diet. 

## 2. Materials and Methods

This study explored the impacts of consumer perception and food choice on women who participated in purposively selected artisan cooperatives. In the qualitative paradigm, phenomenological studies explore lived experiences and seek to understand a person’s experience from their point of view [45,46]. Cooperatives were purposively selected from gatekeepers who were Artisan Cooperative Chief Entrepreneur Founders in sub-Saharan Africa. Criteria for selection were (a) must be an artisan cooperative, (b) that focused on women’s empowerment, and (c) located in areas with reliance on tourism [17]. Two of the cooperatives were in Kigali, Rwanda, while the other cooperative was in Mombasa, Kenya. In total, four separate groups (2 groups at 1 cooperative) of women participated in activities facilitated through Rapid Rural Appraisal Techniques (RRA). The RRA technique is a method used to generate indigenous knowledge in a small amount of time [17]. It is a qualitative methodology conducted by multidisciplinary teams most utilized in assessing problems within agricultural research and development [47]. Different disciplinary perspectives and methods, such as semi-structured interviews alongside hands-on data-gathering activities, make RRA uniquely situated for international development projects [17]. One aim of RRA is to handle the diverse complexities of different individuals and cultures. RRA is often used in agricultural development to rapidly assess rural communities’ needs and evaluate what can be done to improve current systems [48]. At each site, different activities were facilitated with the women to understand the impacts they experienced as part of the cooperative. The activities conducted were to measure social and economic impacts on food security and consumer choice. All questions were orally facilitated and translated by a local female partner. The research design allowed flexibility throughout the day as women were working in the cooperative. This study received Institutional Review Board approval from Texas A&M with protocol number IRB2021-0550D. All participants consented to participating in the RRA activities and having their pictures taken during the activities. Participants also indicated that photos could be utilized in the publication of the research. 

### 2.1. Cooperative Context 

Qualitative research is context-dependent and thus not generalizable. The use of rich descriptions and photography of the RRA activities enhanced the triangulation of data sources within and between cooperatives. The first cooperative was Komera Creative, formed to help women impacted by the Rwanda Genocide in 1994. Many of the women were previously selling fruits on the street or participating in other forms of informal work. Komera Creative seeks to help women learn skills that will enable them to create a sustainable career in the artisan industry. Located in Kigali, the capital, the cooperative employs 10–12 women with varying backgrounds. Komera Creative makes beautifully crafted items that incorporate Getinge (African fabric). Five women were included in the Rapid Rural Appraisal conducted at this site. The women who were part of Komera Creative were all single mothers or single adults. Three of the women had lost their husbands during the Rwandan genocide, while the other two were in their late twenties to early thirties.

Umutima Cooperative was originally a women’s group formed to educate women in Naymirambo, an area just outside of downtown Kigali, Rwanda. Many of the women in the area did not know how to read or write. The cooperative then transitioned from solely teaching women how to read and write to teaching them how to sew. The cooperative wanted to provide an opportunity for women’s economic inputs for themselves and their families. Through connections with a woman in Europe, the women were able to expand the cooperative to others. The cooperative now has over 50 women working together to create Getinge-crafted items. The center also hosts various tourist activities like basket weaving, a walking tour, and a cooking class on top of the artisan goods they make. The women vary in age from their mid-twenties to women in their late forties. Women working in Umutima included single unmarried, single mothers, and married women. There were even a few daughters working at the cooperative. Like Komera Creative, there were also a few women who had lost husbands during the Rwandan genocide.

Imani Collective started in a rural village outside of Mombasa, Kenya. Their founder noticed that emphasis was being placed on the children in providing education, yet many of those children came from homes where their mothers struggled to provide for their families. Their founder then gathered some of those single mothers together and taught them how to sew and started the collective. The cooperative has grown from a few women in the village to now employing over 100 artisans in three separate locations within Kenya, two of which were included in the study, Imani Village (IV) and Imani Old Town (IOT). Those in IOT were mostly married mothers seeking additional family income. Those in IV were mainly single mothers, with some married women and young adults. Those in IV were mainly employed in subsistence agriculture or other informal work which did not provide a consistent income. They were of a lower socio-economic status compared to those living in Mombasa. A larger portion of the women in Mombasa practiced Islam, and we will discuss the impacts this had on their diet further in the Results. In total, across all cooperatives, there were 88 participants in the study. 

### 2.2. Data Collection Using Rapid Rural Appraisal 

RRA is a participatory data-gathering technique. Rather than surveys or questionnaires, a workshop with facilitated activities was most appropriate for gathering local knowledge. These techniques are commonly used in communities where individuals may not have a high literacy rate or numeracy proficiency or when participants are not commonly asked for their views (lacking agency, empowerment, or voice). It is important for the researcher to build rapport prior to facilitating an exercise. The primary researcher was at each site for a week to get to know participants and take field notes from observations. This is called prolonged engagement and persistent observation which is a trustworthiness measure for credibility [45].

The first activity was a food security and nutrition activity. A poster with various food groups was presented to the women. Next to those categories was a scale that had numbers associated with them. These numbers represented the number of times a week a participant would purchase or consume those foods found in that food group. Participants were asked about the time before they were a part of the cooperative; more specifically, they were asked how many times they would consume or purchase any of the food items found within that food group. They would then place a red sticker on the poster in the section that they deemed matched with their behaviors. For example, if a woman felt that she only consumed fruit twice a week before being a part of the cooperative, she would place the red sticker on the scale above the number two. They then repeated the same activity, but the questions were slightly changed to ask what their current behaviors were with consumption and purchasing of that food group since joining the cooperative. They then placed a blue sticker in the corresponding area consistent with their current behaviors in food consumption. The different food groups that the participants were asked about included grains, vegetables, fruits, milk, animal protein, and sugar. 

A priority matrix activity was facilitated informally while the women were working to understand what issues they faced and to determine priorities. Participants were asked what they felt were some of the issues they dealt with before being a part of the cooperative. After the participants discussed this and a list was created, they were asked which issue was the highest priority. For example, if my list had five items, I would ask between items one and two, which was a bigger issue, and then repeat that process by going through all the items in reference to item one. We would continue the process until each item had been compared to all items on the list. Then, the item’s frequency would determine which items were of most importance. The entire process was then repeated with the participants concerning what they felt were the current issues they were facing now since being a part of the cooperative. 

We also asked the women questions in small groups. These were informal interviews with questions designed to understand the impacts of being a part of the cooperative. These questions were follow-up questions to the diet and nutrition activity. We sought to understand why they indicated different trends and changes from pre-cooperative to current. Many of the participants were already working at their machines or tables with a small group of five to seven women, so these natural groups were utilized. These focus groups were effective and did not distract from their productivity as artisans. 

### 2.3. Data Analysis and Trustworthiness Measures

The data were analyzed using a constant comparative approach [49]. Although commonly utilized in other qualitative approaches, this methodology was initially employed for emerging grounded theory. The purpose of comparison and contrast, according to [47], is to “discern conceptual similarities, to improve the discriminative power of categories, and to discover patterns” (p. 96). The primary researcher read over field notes, data from the posters, and other observations and discussions on site. Open coding was used initially to place similar units of data into emerging categories, such as food availability or cooking methods. Next, the categories were compared using axial coding, such as the impact of food availability on cooking methods between categories. Lastly, through selective coding, comparisons across data units were organized into themes and sub-themes, such as cooking methods and food availability in the village vs. urban locations, to determine patterns [49]. 

Each RRA activity was conducted by a facilitator. The women artisans knew and could speak the local language. The researchers asked follow-up questions throughout each activity to understand why the women answered a particular way. These follow-up questions acted as opportunities for members to check and clarify items shared by the participants [45]. The researchers observed the activities of the RRA and only interjected for greater connections between participants to gain trust [50]. The RRA methods provided a visual representation of changes to the women in the cooperative to enhance understanding of individual and collective empowerment. Thus, the “counts” were included to summarize these changes and elicit more responses from participants. These frequencies show local importance and not statistical significance. We did not include all photographic evidence of the RRA activity but compiled the frequency counts into tables for our findings. To preserve dependability and confirmability, we kept a reflective and methodological record throughout the procedure [45]. 

At each cooperative, there were three activities (the food activity, the priority matrix, and interviews) as well as photographs, cooperative documents, and field notes with observations. Each source of data was given a code and included in an audit trail that connected to the emerging themes for triangulation. Peer debriefing memos were created for the emerging themes with audit indications for the original source of data for a peer debriefing with the local leaders and facilitators, as well as the research team. 

This study is limited to the three artisan cooperatives (context-dependent). The intent of qualitative research is not to generalize but to provide a rich description of the women’s experiences in an artisan cooperative. These descriptions provide enough detail for the reader to determine if the findings are transferable. Multiple sources of data were collected, peer debriefing with leaders/facilitators was conducted, and records of all processes were kept, ensuring truth value. The words and photos represent the voices of the participants and not statistical data.

## 3. Results

The women in this study experienced changes due to their participation in their respective cooperatives. While each Rapid Rural Appraisal (RRA) activity measured various aspects of the cooperative’s impact upon their lives, empowerment cannot be defined in only one area. Compelling stories of changes in diet, finances, family dynamics, community, concerns, daily life, future goals, and changes in self-esteem emerged as themes regarding the empowerment of these women. The RRA techniques provided an environment to capture responses within the community. Discussion and participation in matrices were dialogical with activity and translation. Thus, no respondent code is given for statements. Photographs were used to demonstrate the interactive data collection related to these topics where changes were identified. 

### Change in Diet 

The women who took part in this study were of various backgrounds, origins, tribes, and races. This impacted how they applied the adjustments they made due to joining the cooperative. Many women were able to enjoy the standard three meals a day since they were part of a cooperative and had a steadier income. For those who were working, some of the cooperatives offered morning tea or lunch. These changes offered the women even more confidence to try foods they had previously refrained from purchasing, as stated by the women who had the freedom to do so. Women spoke of the independence that came with a steady income. There was no fear of where the next meal would come from. They now dared to buy sugar or go to the supermarket and purchase items they would have previously never felt bold enough to purchase before. Despite having more money and knowing more about healthy eating (many cooperatives offer instruction on a balanced diet), women did not always eat healthier. Sugar was something rarely used across all cooperatives from Rwanda to Kenya. Many women noted that they would spread it out over multiple days or weeks in their morning porridge or tea if they did have sugar (Figure 1 from the Diet Activity). Many women now consume sugar seven days a week, and one woman excitedly shared how she can now have more than one spoonful of sugar in her morning tea. 

Figure 2 demonstrates the participatory approach that RRA can provide in gathering data within communities. Using a typical survey approach is limited in situations like this because many of the women did not read or write. Culturally, it was acceptable with only women present to gather this data as a social activity. The engagement in this approach is insightful in determining how these social enterprises develop educational programming to assist in decision-making and consumer choice that was not available to them prior to participating in the cooperative.

Kenya and Rwanda both lie along the equator, which makes growing both vegetables and fruit easier. Women in Rwanda said adults did not consume fruits that often. One participant said that they were told babies needed fruit, so they would give fruit to children and not eat it themselves. They were able to gain more knowledge about the value of fruit in their diet after joining the cooperative. The average fruit consumption before being a part of the cooperative across all groups ranged from less than half a day (0.4) a week to 3.2 days a week. After being a part of the cooperative, it changed from 3.2 to 6.2 days a week. The greatest change among all groups from before to after was experienced by Komera Creative, where their change in consumption of fruit increased by three days a week (see Figure 3). Multiple individuals expressed that they felt they now had the option of purchasing fruit for the whole family. One participant from the Imani’s Village location described how when she saw fruit vendors with their carts walking around, she now felt confident in being able to buy fruit if she wanted. In the village, common fruits consumed were banana, mango, or papaya.

The consumption and availability of fruit options change with the seasons. At Imani’s location in the village, women shared how it was currently mango season. They were always eating mangoes because mango trees surrounded them, and people would be giving away mangos for free. Culture impacted the perception of how many of these women changed their fruit consumption. Now, participants see fruit as something vital to their diet, not just their children. 

In Imani’s village location, many women grew vegetables in their gardens, or they could easily buy them because they were cheap and grown locally year-round. The average consumption before cooperative participation was 6.2 days a week at Imani’s village location. But after being a part of the cooperative, it dropped to 3.4 days a week. Even with training on healthy diets, why would women decrease their consumption of vegetables? The women in the village noted that before, they were “forced” to buy inexpensive vegetables for meals every day instead of meat, plant protein, or other more expensive alternatives. Vegetables were “cheap” and there was a discussion about how a steady income allowed them to buy other food items. They could now buy meat or other expensive grains or fruits for meals. Some of the most readily available vegetables to most of the women were green beans, carrot, eggplant, spinach, onion, and scumawhichi (a leafy plant). While still using their agency to choose food that had previously been unavailable to them, women were making their own choices rather than being dictated to by what was cheap and available to them. Those in the village still lacked a wider availability of different foods compared to participants living in more urban areas.

Figure 4 provides a specific view of the village diet before and after participating in the cooperative. With increased wages, the women were eating fewer vegetables because they had money to purchase foods that were considered luxurious before. 

Another change in diet was with milk consumption. In Rwanda, milk is featured on the national currency, and milk bars (like a cocktail bar, but with milk instead) are found throughout every town. Women in Umutima consumed milk 3.7 times a week before being a part of the cooperative, but it changed to 5.4 after participating in the cooperative. The women’s group in Komera had lactose intolerant individuals, so they avoided milk. Yet both groups mentioned that in Rwanda, you knew you were destitute if you could not afford to buy milk even once a week for morning or afternoon tea. It was common to have milk mixed with their morning tea, even in Kenya. In Imani Old Town, women could drink milk every day because it was provided to them. At the same time, some women in Imani village consumed more milk because they had a cow on their farm or could purchase a cow after they started working at the cooperative. Interestingly, one of the women who put six for the number of times she had milk during a week was questioned by all the other women on why she would not put seven. Some of the women were even telling her to grab her sticker and move it to the seven because Imani has tea every day, which includes milk. She told the other women that she chose to not take milk on her day off, so she could answer six if she wanted to. Compared to other questions, the women in Imani’s Old Town were quick to answer how many times they drank milk a week since joining the cooperative, and due to peer pressure, very few of them deferred from putting their sticker under the seven. The amount of milk consumption also increased sugar consumption. Having daily tea with milk meant women added sugar to sweeten the drink. 

Every group of women laughed when asked to put the sticker on how many times they consumed or bought meat in a week. Some women even asked if there was a spot for once a month or a couple of times a year. Meat consumption was reserved for special holidays and was still seen as expensive. Particularly for those who practiced Islam, meat was eaten sparingly and was mainly eaten at times of holidays such as Eid. Women were only asked about animal protein and not plant protein for this activity. It is common to have beans or green grams or other legumes and plant proteins with grains or vegetables for meals. Before the cooperative, the average meat consumption was less than ½ a day to 1.76 days a week across all groups. Current consumption did not change much, ranging from 2.4 to 3.04 days a week. Daily meat consumption was something still seen as reserved for the rich. Grains and vegetables were cheap and easy to add into their daily diet, while meat was reserved for extraordinary events or a few times a week. A few women smiled at the others while they confidently put their meat stickers under five, six, or seven times a week. Even if they could afford animal protein, other forms of protein were still more common within each culture and country for daily meals. In Rwanda, it is very common to have beans with rice or some other grain or starch for meals. In Kenya, beans and other legumes are also readily used for lunches. When the researcher was invited for meals offered at the cooperatives, plant proteins with rice, chapati, potato, or some other grain were typical. Types of meat available to women also impacted these choices. In the village, women had access to most animal proteins, including fish, because they were located next to the ocean. Women working in IOT had the same proteins but would avoid different animal proteins for religious reasons. Women in Rwanda, though, usually consumed beef, goat, chicken, or lamb. Meat was something the women still saw as being reserved for special occasions only.

Another important factor in diet and nutrition were family dynamics. Many women expressed in the small group interviews how an increased income allowed them the ability to choose different foods, but family still dictated those choices. Husbands would ask for the money they had earned, or children would ask for foods they preferred. Extended family members learned of their stable income and asked for financial assistance at times. Women shared how they felt pressure to give their income to others, thus limiting the amount of money available for food.

The opinions and preferences of food in the city and the village were quite different. People who live in cities have access to a wider variety of foods. People in the village, however, had less variety but more readily available locally sourced food, such as vegetables, as previously mentioned. Those in the city had an average consumption across all food groups higher in the “before” diet activity than the villages. When considering issues with transporting food and lack of refrigeration, villages become food insecure during droughts, floods, or other national disasters. However, cities may have more variety available, but at higher costs. Figure 5 shows the changes in diet before and after in rural vs. urban communities.

Due to cultural nuances and locations, we have included a comparison of all three cooperatives in six food categories. Figure 6 shows the diet changes before and after across cooperatives. 

Before the cooperative, women typically prepared food based upon the expense of fuel. The cheapest way to cook food was using a charcoal stove, then using paraffin and natural gas would be the most expensive. Cooking with a charcoal stove versus a gas stove impacted how and what food was prepared. In IOT, most women expressed how they were now able to cook with a gas stove after joining the cooperative. Some women in Imani Village were given gas stoves or ovens as prizes during a party provided by the cooperative. Different foods cook faster and are better adapted for the gas stove, while others are not. Relying on a charcoal stove requires longer cooking time, but now many women cook with all three stoves at the same time! Some preferred the way matooke (green, starchy banana) was cooked on charcoal, but they could easily prepare rice on the gas stove. Women now choose which foods to prepare because they can use different stoves at varying times.

## 4. Discussion

We see women’s diets changed through their participation in a cooperative. The social networks established within the artisan cooperatives provided women greater access to resources, information, and social mobility to change their diet [15,31]. Being a part of the cooperative has provided many women with decent work and the ability to move upward in their economic mobility because of their increased level of social capital. This decent wage, which aligns with SDG 8, has also allowed them to reduce poverty and hunger within their own households. Like [19], women felt greater levels of empowerment, which contributed positively towards their own agency and food choices. However, some women spoke about how the steady income was shared with their husbands or family members instead, so some women were limited in their direct influences on diet in those situations. 

Women working in the village locations experienced positive impacts on their household food security because of the social networks provided by the cooperative [35,37,38]. The women’s courage and choices were broadened through their participation. Many mentioned having greater freedom to choose what they wanted to eat versus finding cheap food to maintain daily life and survive. Agency or the ability to choose seemed to be the overarching theme related to how the women expressed changes in their diets [19].

Women were given training on proper nutrition and provided meals while working at the cooperative. The ability of the women to gain knowledge about improved diets reduced the barriers towards improving their household’s diet [6]. When given a choice with a decent wage, they wanted more sugar. Women in Imani Village indicated they no longer needed to eat local vegetables (free or cheap to buy). Agency for these women meant even with increased knowledge of the benefits of vegetables, the ability to choose what they wanted empowered them [19]. This is a notable example for researchers and practitioners to find ways to both develop culturally viable products and communicate culturally cognizant nutrition guidelines.

## 5. Conclusions

Leaders of social enterprises who want to improve livelihoods and food security need to realize that shifts in diet choices (such as increasing sugar and fruit) are symbols of economic success. Knowing that vegetables should still be consumed at a higher level may need a more sensitive approach related to healthy lifestyles. Providing other healthy, culturally viable alternatives may increase nutrients and vitamins within the new consumer choice paradigm. Investing time and research into understanding the role of local plants and foods that are culturally viable can be a way to help improve household food security rather than coming from a Western perspective [22]. Understanding the differences in diet and food availability in rural versus urban locations is an important consideration. What is available to women in Mombasa because of their location on the coast compared to women in Rwanda living in a landlocked country would vary and need specialized workshops. In Mombasa, they have access to other animal proteins like fish compared to Kigali, Rwanda, where the closest body of water is multiple hours away. Even with increased income, the availability of food remains a barrier towards improving household food security and diet [6]. Understanding attitudes toward current diets and trends with cultures can act as predictors for food choices [16,18,26]. This study was focused more regionally in East Africa and did not have statistical significance, but with the data and methods shared can still be transferable to other similar contexts as deemed by the reader. 

This may be an opportunity for practitioners to provide short-term experiential learning opportunities such as cooking classes on culturally viable, healthier alternative food items or increased knowledge of food items [25]. There also needs to be a greater understanding in each cooperative location of food availability within cultural mores (i.e., meats for festivals). The women in the rural areas indicated a lower availably of options for food choices compared to those in urban areas. This understanding will provide practitioners and researchers knowledge on what nutrition programs and plans can be implemented for women to access locally grown or proximity food choices.

Further research should examine social benefits and connections found within artisan cooperatives that lead towards greater economic gain with the goal of improved household food security. Agencies and government organizations should seek to partner with these artisan groups to provide sustainable development to promote the women’s agency while giving them the knowledge to make informed decisions as consumers, thus building on systems that contribute towards reducing gender inequality. 

## Figures and Tables

**Figure 1 foods-12-03956-f001:**
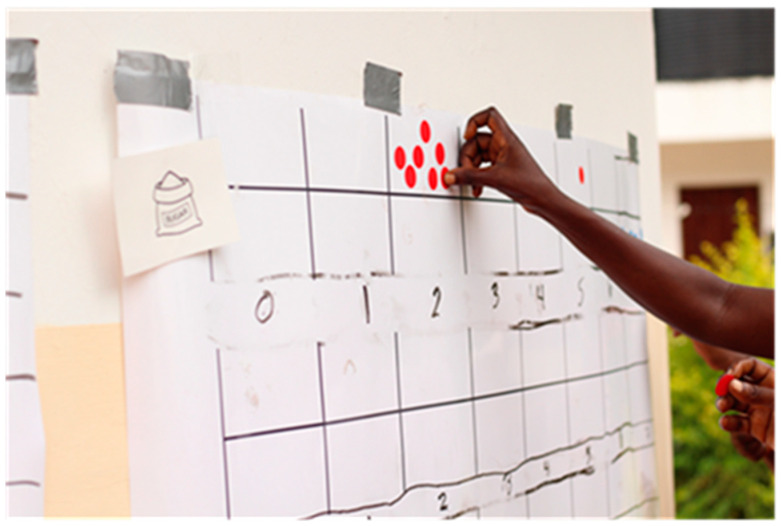
Women indicating sugar before cooperative consumption.

**Figure 2 foods-12-03956-f002:**
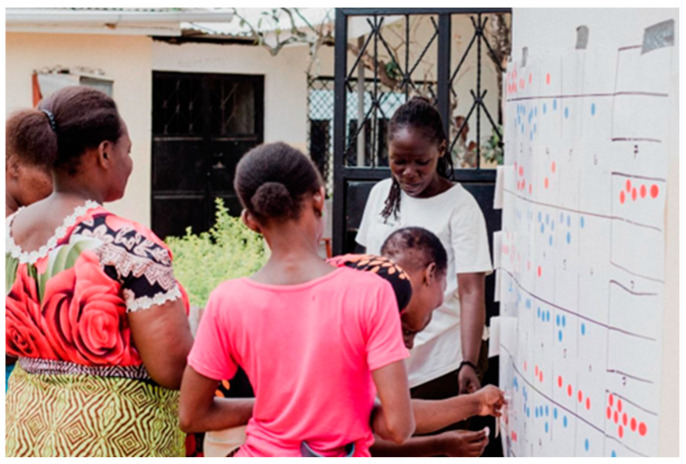
Women participating in the diet activity while discussing the changes in their diet with the group. Red dots are for before the cooperative, and blue dots represent after joining the cooperative.

**Figure 3 foods-12-03956-f003:**
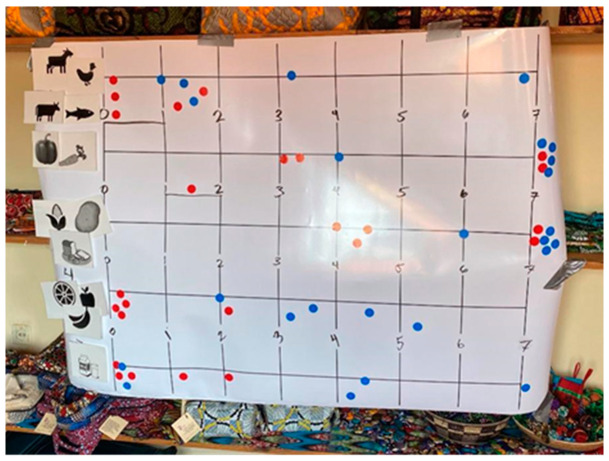
The results of the diet activity at Komera Creative using Rapid Rural Appraisal.

**Figure 4 foods-12-03956-f004:**
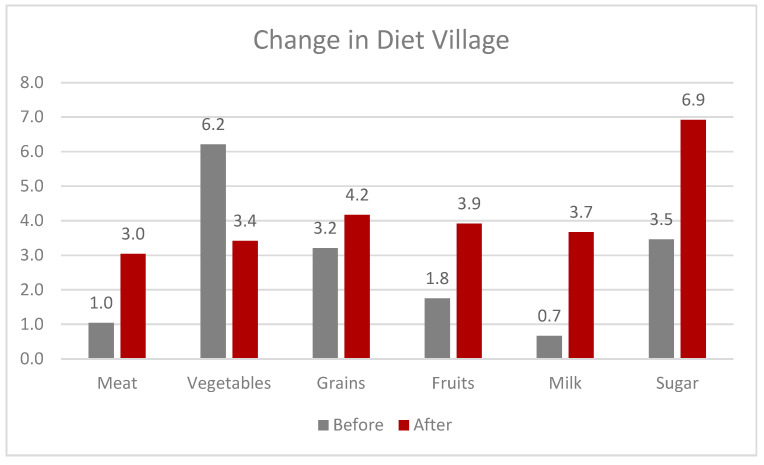
The figure shows the overall changes in the diet of the women working in the village.

**Figure 5 foods-12-03956-f005:**
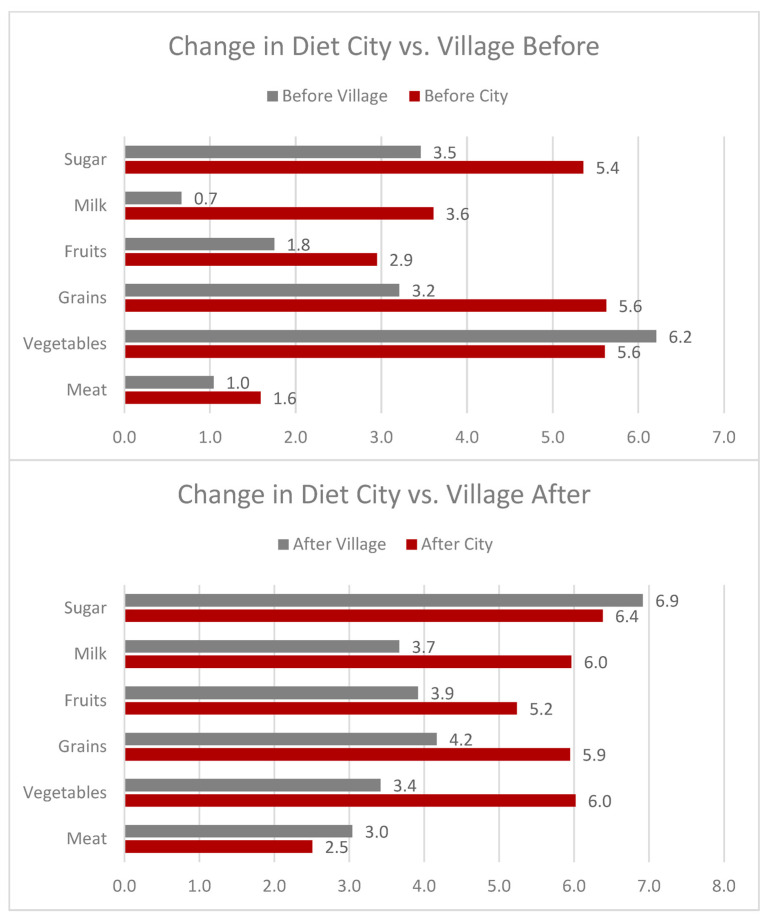
These figures show the before and after difference in consumption of different food items between the city and village areas.

**Figure 6 foods-12-03956-f006:**
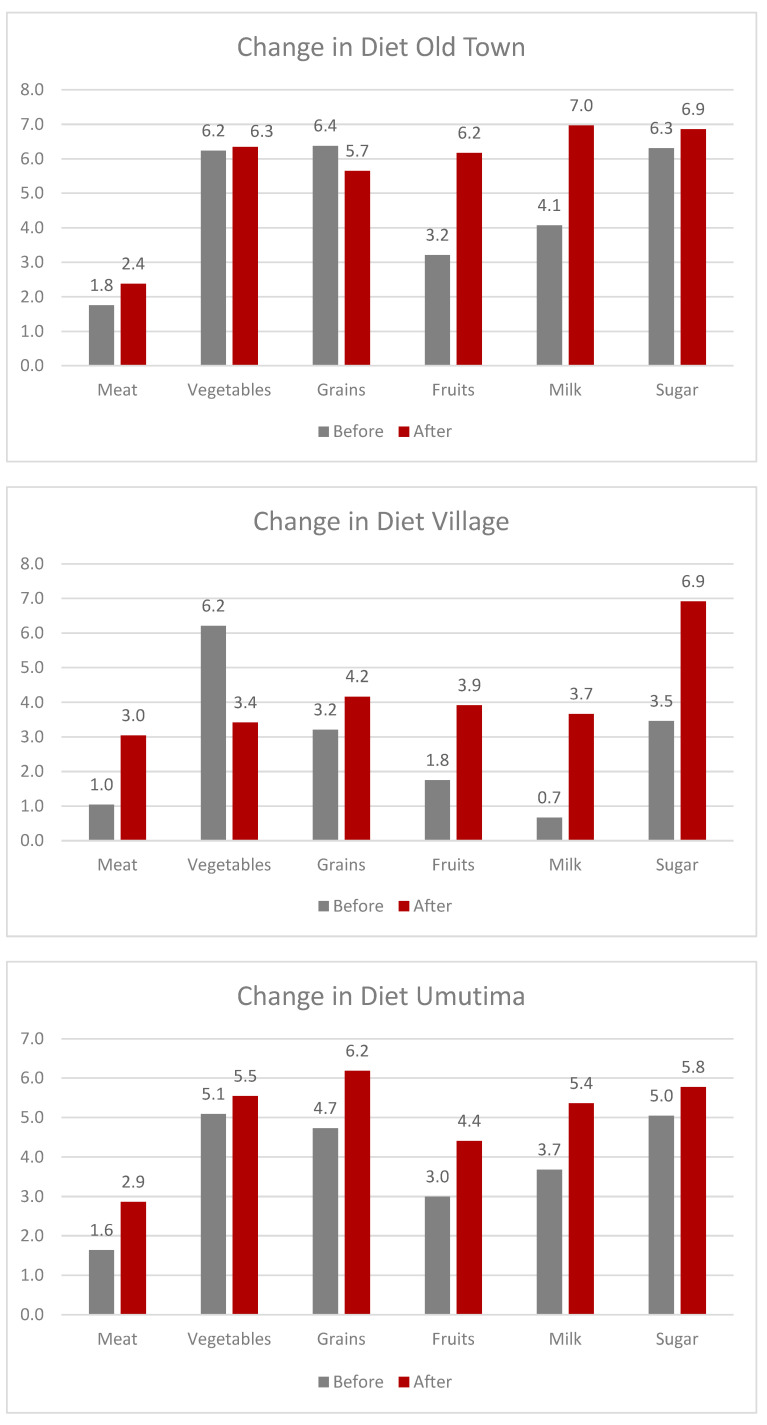
These figures detail the changes in diet within all the cooperatives’ locations.

## Data Availability

Data is contained within the article.

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
