# Peer review of "How Does an Artisan Cooperative Impact Food Perception and Consumer Behaviors? A Rapid Rural Appraisal of Women in East Africa"

_foods, 2023, doi:10.3390/foods12213956_

Round 1

Reviewer 1 Report

Comments and Suggestions for Authors

1.Some deficiencies and areas for improvement in the Abstract could be identified as follows:

(1)The Abstract should clearly state the research objectives or research questions being addressed in the study. It is important to highlight what specific aspects of social entrepreneurship, consumer perception, and food choice were examined.

(2)The Abstract briefly mentions the adoption of a phenomenological perspective and data collection using Rapid Rural Appraisal methodologies, but it lacks details regarding the sample size, data collection methods, and analysis techniques employed. Providing these details would enhance the clarity and provide a better understanding of the study design.

(3)The Abstract mentions that women did not always make healthier food choices despite having more information and income. However, it does not provide specific analysis findings or quantitative/qualitative evidence to support this statement. Including key findings or significant results would strengthen the Abstract's content.

(4)The Abstract concludes abruptly without a summary or concluding statement. It is crucial to provide a brief summary of the implications or significance of the study's findings within the Abstract itself.

2.Some deficiencies and areas for improvement in the Introduction could be identified as follows:

(1)The introduction lacks clarity regarding its central research focus. It introduces a wide range of topics, from consumer choices and development goals to cooperatives and social capital theory. It is important to clearly state the research question or hypothesis and the main objective of the study.

(2)The introduction lacks a clear organizational structure. It jumps between topics without a smooth transition, making it challenging for readers to follow the logical flow of the argument.

(3)The introduction should clearly identify the research gap or problem that this study aims to address. What is missing from the existing literature on which this research contributes?

(4)While the introduction mentions the study's alignment with Sustainable Development Goals (SDGs), it does not discuss the scope and limitations of the research. What are the boundaries of this study and the potential limitations that might affect the results?

(5)The Introduction should synthesize and summarize the key findings and debates in the existing literature related to the research topic. This will help to establish the context of this study.

3.The Materials and Methods section of this study has several deficiencies and areas that could be improved.

(1)This section mentions that cooperatives were purposively selected based on a previous study, but it does not provide any information about how these cooperatives were chosen or why they were selected. A clear justification for the selection criteria and discussion of potential biases or limitations related to this sampling approach should be included.

(2)While the section mentions the use of RRA techniques, it does not describe them in detail. Readers should be provided with a comprehensive explanation of how the data were collected, including the specific tools or instruments used, the sequence of data collection activities, and the duration of data collection at each site.

(3)The description of the data analysis was brief and lacked clarity. While it mentions the use of a constant comparative approach, it does not provide details on how this approach was applied to the collected data. A more comprehensive explanation of the data analysis process, including the coding and thematic analysis, is required.

(4)While the section mentions data triangulation, it does not elaborate how multiple data sources were used to corroborate the findings. Providing examples of how different data collection methods were compared and integrated would enhance the credibility of this study.

(5)This section lacks clear subheadings or a structured format, making it challenging for readers to follow the flow of the research methodology. A well-organized structure with subheadings for each key aspect of the methodology would improve readability.

4. The Results section of this study had several limitations. The following are some key points to consider:

(1)There is no mention of statistical analyses or tests to determine the statistical significance of the observed changes in diet and other aspects. Without statistical analysis, it is challenging to determine whether the observed changes are statistically meaningful or simply due to chance.

(2)The study discusses the experiences of women in cooperatives in specific locations (Kenya and Rwanda) and may not be generalizable to other contexts or populations. The absence of a broader discussion of the external validity of the findings limits the applicability of the study beyond the studied communities.

(3)The section lacks consistency in reporting the results. For example, some figures are referenced but not provided in the text, and there is a lack of clarity regarding how they relate to the findings. This makes it difficult for the readers to interpret the data.

(4)While the study mentions cultural factors influencing dietary changes, it does not provide a deep exploration of these cultural factors or how they may interact with empowerment. A more in-depth analysis of the cultural context would enrich this discussion.

(5)While the study presents findings from different cooperatives and locations, there is limited synthesis and comparison of these findings. A more in-depth discussion of the similarities and differences between cooperatives would enhance this analysis.

5.The Discussion section of this study had several limitations. The following are some key points to consider:

(1)The discussion briefly mentions a study from the introduction, but does not effectively compare or contextualize the findings of the current study with existing research. It is important to provide a more comprehensive literature review and discuss how the current findings align or differ from previous studies in the field.

(2)The discussion tends to make broad statements about women's experiences within artisan cooperatives, without considering potential variations across different contexts or regions. Acknowledging the potential heterogeneity of experiences and outcomes is essential.

(3)The discussion does not adequately address the limitations of the study, such as the sample size, data collection methods, or potential biases. Acknowledging and discussing these limitations are crucial for a thorough academic review.

(4)While the discussion mentions the importance of culturally viable products and nutrition guidelines, it does not provide concrete examples or strategies for achieving cultural sensitivity in nutrition programs. Academic discussions should offer practical insights and recommendations based on the research findings.

(5)While the discussion briefly mentions the need for agencies and government organizations to partner with artisan groups, it does not elaborate on specific policy implications or recommendations for sustainable development. Providing actionable policy recommendations can enhance the practical relevance of this study.

Author Response

We really appreciated your detailed feedback. Your recommendations helped us to focus on critical aspects to improve our scholarship for both science and practice. We have attached how and where in the document we addressed your edits and recommendations. 

Reviewer #1

The authorship team appreciates your detailed review and feedback to help us improve our scholarship. We addressed each piece of your feedback and edits in blue below including the new page and line(s) locations in the revised manuscript.

  1. Some deficiencies and areas for improvement in the Abstract could be identified as follows:

(1)The Abstract should clearly state the research objectives or research questions being addressed in the study. It is important to highlight what specific aspects of social entrepreneurship, consumer perception, and food choice were examined.

We better clarified what the research objectives were on line 11 and 12.

(2)The Abstract briefly mentions the adoption of a phenomenological perspective and data collection using Rapid Rural Appraisal methodologies, but it lacks details regarding the sample size, data collection methods, and analysis techniques employed. Providing these details would enhance the clarity and provide a better understanding of the study design.

Our authorship team added further clarity on the RRA methods starting on line 17.

(3)The Abstract mentions that women did not always make healthier food choices despite having more information and income. However, it does not provide specific analysis findings or quantitative/qualitative evidence to support this statement. Including key findings or significant results would strengthen the Abstract's content.

We included a few more key findings starting on line 27.

(4)The Abstract concludes abruptly without a summary or concluding statement. It is crucial to provide a brief summary of the implications or significance of the study's findings within the Abstract itself.

Authors added better concluding summary starting on line 35.

  1. Some deficiencies and areas for improvement in the Introduction could be identified as follows:

(1)The introduction lacks clarity regarding its central research focus. It introduces a wide range of topics, from consumer choices and development goals to cooperatives and social capital theory. It is important to clearly state the research question or hypothesis and the main objective of the study.

We revised and adjusted the Introduction for overall study with better clarity and research objectives. This starts on line 37.

(2)The introduction lacks a clear organizational structure. It jumps between topics without a smooth transition, making it challenging for readers to follow the logical flow of the argument.

We adjusted the transition and had a more specific introduction and then moved to literature review.

(3)The introduction should clearly identify the research gap or problem that this study aims to address. What is missing from the existing literature on which this research contributes?

We adjusted and more clearly stated what the research gap and problem was which can be found within the introduction and more specifically in lines 60 to 75.

(4)While the introduction mentions the study's alignment with Sustainable Development Goals (SDGs), it does not discuss the scope and limitations of the research. What are the boundaries of this study and the potential limitations that might affect the results?

We clarified this later not within the Introduction but within the methods section starting on page 4.

(5)The Introduction should synthesize and summarize the key findings and debates in the existing literature related to the research topic. This will help to establish the context of this study.

Had better summary of key findings within Introduction that started on page 2. This helped lead better into the literature review.

  1. The Materials and Methods section of this study has several deficiencies and areas that could be improved.

(1)This section mentions that cooperatives were purposively selected based on a previous study, but it does not provide any information about how these cooperatives were chosen or why they were selected. A clear justification for the selection criteria and discussion of potential biases or limitations related to this sampling approach should be included.

Authors included better understanding of selection criteria on page 4 starting on line 176.

(2)While the section mentions the use of RRA techniques, it does not describe them in detail. Readers should be provided with a comprehensive explanation of how the data were collected, including the specific tools or instruments used, the sequence of data collection activities, and the duration of data collection at each site.

We described already what are activities were and how they were conducted within the methods section which started on line 246. We included more information about RRA methods starting on line 182.

(3)The description of the data analysis was brief and lacked clarity. While it mentions the use of a constant comparative approach, it does not provide details on how this approach was applied to the collected data. A more comprehensive explanation of the data analysis process, including the coding and thematic analysis, is required.

The methodology was expanded on how we used the constant comparative method within our results on page 6 starting on line 292.

(4)While the section mentions data triangulation, it does not elaborate how multiple data sources were used to corroborate the findings. Providing examples of how different data collection methods were compared and integrated would enhance the credibility of this study.

We explained more about triangulation and what other sources of data were included within the data analysis section, starting on lines 306 and 307.

(5)This section lacks clear subheadings or a structured format, making it challenging for readers to follow the flow of the research methodology. A well-organized structure with subheadings for each key aspect of the methodology would improve readability.

We added a few more subheadings to better guide the reader and changed the wording of a few of the headings.

  1. The Results section of this study had several limitations. The following are some key points to consider:

(1)There is no mention of statistical analyses or tests to determine the statistical significance of the observed changes in diet and other aspects. Without statistical analysis, it is challenging to determine whether the observed changes are statistically meaningful or simply due to chance.

Our study was purely qualitative and no statistical data was analyzed. We just reported frequency counts and displayed them in tables instead of showing all the photos of each chart. We indicated this within our methods section starting on page 4 line 188, page 6 line 265, and page 7 line 201.

(2)The study discusses the experiences of women in cooperatives in specific locations (Kenya and Rwanda) and may not be generalizable to other contexts or populations. The absence of a broader discussion of the external validity of the findings limits the applicability of the study beyond the studied communities.

We shared further about transferability as one of the ways readers can apply what was done in this study to their context. This was discussed within the methods section on page 7 starting on line 304.

(3)The section lacks consistency in reporting the results. For example, some figures are referenced but not provided in the text, and there is a lack of clarity regarding how they relate to the findings. This makes it difficult for the readers to interpret the data.

We made sure all figures are referenced correctly within the text. We felt that the figures add to guiding the reader specifically in understanding how RRA methods were used in this context.

(4)While the study mentions cultural factors influencing dietary changes, it does not provide a deep exploration of these cultural factors or how they may interact with empowerment. A more in-depth analysis of the cultural context would enrich this discussion.

Authors provided more examples within the results section cultural items. For example on page 10 starting on line 412.

(5)While the study presents findings from different cooperatives and locations, there is limited synthesis and comparison of these findings. A more in-depth discussion of the similarities and differences between cooperatives would enhance this analysis.

We shared more about the differences and similarities between cooperatives within the results section. For example on page 11 starting on line 460, we added more similarities and differences.

5.The Discussion section of this study had several limitations. The following are some key points to consider:

(1)The discussion briefly mentions a study from the introduction, but does not effectively compare or contextualize the findings of the current study with existing research. It is important to provide a more comprehensive literature review and discuss how the current findings align or differ from previous studies in the field.

We revised this and included the correction citation this is on page 14 line 553.

(2)The discussion tends to make broad statements about women's experiences within artisan cooperatives, without considering potential variations across different contexts or regions. Acknowledging the potential heterogeneity of experiences and outcomes is essential.

Authors discussed within the results and conclusions the varying results between regions and urban and rural (ex. page 14 line 555).

(3)The discussion does not adequately address the limitations of the study, such as the sample size, data collection methods, or potential biases. Acknowledging and discussing these limitations are crucial for a thorough academic review.

Authors acknowledge limitations earlier on and added a line within Conclusions on page 15 lines 548-551.

(4)While the discussion mentions the importance of culturally viable products and nutrition guidelines, it does not provide concrete examples or strategies for achieving cultural sensitivity in nutrition programs. Academic discussions should offer practical insights and recommendations based on the research findings.

We provided a few more suggestions and discussion on culturally viable products starting on page 15 on line 566.

(5)While the discussion briefly mentions the need for agencies and government organizations to partner with artisan groups, it does not elaborate on specific policy implications or recommendations for sustainable development. Providing actionable policy recommendations can enhance the practical relevance of this study.

We added more to last paragraph on lines 563 – 567 (page 15) to discuss what government agencies should do based on our data.

Reviewer 2 Report

Comments and Suggestions for Authors

The manuscript reports the influence of artisan cooperation on consumer perception and food choice in women in East Africa.

My main concern related to this manuscript is the ethical aspect of this investigation. Does some research ethics committee approve this research?

Do authors have permission from people to publish picture 2?

Additional points:

Please provide at least a short explanation for each mentioned SDG.
Since the abbreviation for Social Capital Theory was defined, it should be used throughout the further text.
The methodology section should provide data on the total number and some socio-demographic characteristics of participants in each involved region.
Please add data analysis as a separate sub-section. How were the results expressed? What do mean values presented in figures? Why is there no statistical analysis of data?
Add the conclusion with further directions as a separate section.

Comments on the Quality of English Language

Some editing for English is required.

The manuscript would benefit from professional editing.

Author Response

Thank you for the time you devoted to reviewing and helping us improve our scholarship. We have attached how and where in the document we addressed your edits and recommendations. 

The authorship team appreciates your detailed review and feedback to help us improve our scholarship. We addressed each piece of your feedback and edits in blue below including the new page and line(s) locations in the revised manuscript.

  1. To address this respective feedback, “My main concern related to this manuscript is the ethical aspect of this investigation. Does some research ethics committee approve this research?”

We received Texas A&M University Institutional Review Board (IRB) approval for protocol number IRB2021-0550D.

  1. The reviewer indicated, “Do authors have permission from people to publish picture 2?

We received permission to publish picture 2 from the IRB Review Board and participants and have added that permission on page 5 lines 197 to 200.

  1. Reviewer’s Additional points:  

a. Please provide at least a short explanation for each mentioned SDG.            

We described each SDG and this occurred on page 2 lines 90 to 110.

b. Since the abbreviation for Social Capital Theory was defined, it should be used throughout the further text.

We revised and used SCT throughout the entire document.

c. The methodology section should provide data on the total number and some socio-demographic characteristics of participants in each involved region.

We included more in within the methods section and results about the demographics of participants. For example, on page 5 lines 211 to 214.

d. Please add data analysis as a separate sub-section. How were the results expressed? What do mean values presented in figures? Why is there no statistical analysis of data?

We clarified this within our methods and results section. Our study was purely qualitative, and the frequency counts and tables used were only to elicit information and responses from participants. No statistical analysis was conducted on the data.

e. Add the conclusion with further directions as a separate section.

We added Conclusion as its own section on page 14 line 544.

Thank you again for the time you dedicated to providing us detailed feedback to improve our scholarship.

Round 2

Reviewer 1 Report

Comments and Suggestions for Authors

The text has been thoroughly revised by incorporating feedback provided by the reviewer. The authors have successfully addressed my comments and suggestions, thereby enhancing both the quality and clarity of the manuscript. I am now in a position to recommend acceptance.

Author Response

Thanks for the reviews to help us improve our scholarship and the dissemination to both scientists and practitioners. 

Reviewer 2 Report

Comments and Suggestions for Authors

The authors made efforts to address the concerns of reviewers. The current version of the manuscript is improved compared to the submitted version. However, the quality of the Figures should be improved. Uniform the decimal numbers. The titles of the figures should be rephrased to be more concise.

Author Response

We appreciate the recommendations to improve the Figures and Illustrations. We have addressed those and the detail of the revisions are located in the attachment. 

The authors made efforts to address the concerns of reviewers. The current version of the manuscript is improved compared to the submitted version. However, the quality of the Figures should be improved. Uniform the decimal numbers. The titles of the figures should be rephrased to be more concise.

We went through and fixed all the figures to have the same decimal numbers and be more consistent across the whole article. Reduced the titles of the figures as well an rephrased some of them for more consistency. Adjusted everything to be the same size too.